# Changes in Bone Marrow Lesions Following Root Repair Surgery Using Modified Mason–Allen Stitches in Medial Meniscus Posterior Root Tears

**DOI:** 10.3390/medicina58111601

**Published:** 2022-11-04

**Authors:** Kyu Sung Chung, Jeong Ku Ha, Jin Seong Kim, Jin Goo Kim

**Affiliations:** 1Department of Orthopedic Surgery and Sports Medical Center, Sports Medical Research Institute, Seoul Paik Hospital, College of Medicine, Inje University, Seoul 04551, Korea; 2Department of Sports Medical Center, Sports Medical Research Institute, Seoul Paik Hospital, College of Medicine, Inje University, Seoul 04551, Korea; 3Department of Orthopedic Surgery and Sports Center, Myong-Ji Hospital, Seoul 10475, Korea

**Keywords:** bone marrow lesion, knee, meniscus, root tear, root repair

## Abstract

*Background and Objectives*: Root repair can prevent osteoarthritis (OA) by restoring hoop tension in medial meniscus posterior root tears (MMPRTs). This study aims to investigate bone marrow edema (BME) lesions known to be associated with OA following MMPRTs. *Methods*: Thirty patients with transtibial pull-out repair were recruited. Subchondral BME lesions were evaluated using magnetic resonance imaging (MRI) at 1-year follow-ups. Participants were categorized into three groups: no change of BME lesions (group one), improved BME lesions (group two) and worsened BME lesions (group three). Clinical scores and radiological outcomes, specifically Kellgren–Lawrence grade, medial joint space width and cartilage grade and meniscal extrusion were evaluated and compared between groups. *Results*: After surgery, twenty-three patients with no BME, three patients with BME lesions on the medial femoral condyle, one patient with BME lesions on the medial tibia plateau and three patients with BME lesions on both were investigated. A total of 20 patients in group one (66.7%) showed no change in BME lesions. In group two, seven patients (23.3%) presented with improved BME lesions. Only three patients (10%) showed worsened BME lesions (group three). Moreover, Lysholm scores and the rate of progression of cartilage grades were significantly worse in group three patients. Meniscal extrusion was significantly reduced in group two, whereas extrusion was significantly progressed in group three. *Conclusions*: Patients with worsened BME lesions showed less favorable outcomes than other patients. A decrease in meniscal extrusion can have a positive effect on BME lesions after root repair.

## 1. Introduction

Bone marrow edema (BME) is defined as an area of low signal intensity on T1-weighted images that is associated with intermediate or high signal intensity findings on T2-weighted images. BME lesions are a risk factor for structural deterioration in knee osteoarthritis and are strongly associated with the progression of osteoarthritis (OA) [1,2,3].

The meniscus is composed of an interconnecting network of collagen fibers, proteoglycans and glycoproteins [4,5]. Collagen fibers stretch under axial-load-increasing internal hoop stress which absorbs and redistributes forces transmitted to the joint [6,7,8,9,10]. Medial meniscus posterior root tears (MMPRTs) are defined as an avulsion injury or radial tear occurring in posterior bone attachment [11,12,13,14,15,16]. MMPRTs can predispose patients to degenerative OA due to the loss of hoop tension and load-sharing ability, which results in unsupportable peak pressures on the knee joint [14]. Root repair can restore the meniscal function; thus, it can prevent or delay osteoarthritic changes [17,18,19]. Root repair shows superior mid- [20] and long-term [21] clinical outcomes when compared with meniscectomy.

One of the main goals of root repair is the prevention and delay of the onset of osteoarthritic changes through the restoration of meniscal hoop tension. If root repair is not effective or successful, BME lesions will worsen following surgery. Several studies of both the clinical and radiological outcomes of root repair have reported favorable results [22,23,24,25,26,27]. However, there is little evidence or investigation into the change of BME lesions following root repair.

It has been shown that greater meniscus extrusion is a significant predictor of the progression of arthritic changes in osteoarthritic knees [18]. Therefore, it seems logical that if meniscus extrusion can be eliminated or reduced, the MMPRT will be successfully repaired, and the chance of subsequent degenerative arthritis will be reduced. However, there are no studies that have investigated the correlation between meniscus extrusion and BME lesions.

As such, this study aims to clarify the treatment effects of root repair through investigating the change of BME lesions following root repair and comparing these with both pre- and post-operative 1-year follow-up results in MMPRTs. In addition, this study aims to investigate the correlation of meniscus extrusion and BME lesions after root repair. We hypothesize that patients with worsened BME lesions will show less-favorable outcomes than other patients and that patients with more reduced meniscal extrusions can demonstrate positive effects on BME lesions after root repair.

## 2. Materials and Methods

### 2.1. Study Population

This study protocol was approved by our institutional review board (PAIK-08-003), and all patients signed an approved written consent form. Between 2017 and 2020, patients who consented underwent magnetic resonance imaging (MRI) scans both prior to surgery and 1-year following surgery. MMPRT was defined as a complete radial tear on the medial meniscus posterior bone. This was diagnosed by MRI scans (Intera Achieva; Philips, Eindhoven, Netherlands) indicated via an absence of an identifiable meniscus or a high-intensity signal replacing the normal dark meniscal signal (ghost sign) in the sagittal plane, a vertical linear defect at the root on the coronal plane, and a radial linear defect at the posterior insertion point in the axial plane [28,29].

The inclusion criteria for the study were: first, MMPRT shown on an MRI scan in a patient with persistent knee pain; second, patients who underwent arthroscopic pull-out fixation by modified Mason–Allen stitches; third, patients with a Kellgren–Lawrence (K-L) score of grade two or less. Study exclusion criteria were: first, patients whose MMPRT was combined with a high tibial osteotomy; second, patients with a concomitant ligament injury; and finally, patients who did not want an MRI scan follow-up 1-year after surgery (Figure 1).

Participants were identified using medical records. Prior to surgery, participants were interviewed in the outpatient department and their clinical histories were reviewed and collated in a database.

### 2.2. Evaluation of Bone Marrow Lesions

The MRI scans were evaluated both prior to surgery and 1-year following surgery. The evaluations of the BME lesions at both the medial femoral condyle (MFC) and medial tibial plateau (MTP) were conducted. Changes of the BME lesions were assessed by comparing the statuses of the BME lesions between MRI scans prior to surgery and 1-year post surgery. Participants were categorized into 3 groups: no change of BME lesions (group 1), improved BME lesions (group 2) and worsened BME lesions (group 3), through comparisons between pre- and post-operative BME lesion statuses (Figure 2 and Figure 3). The cartilage grade in the medial compartment was graded according to the modified Outerbridge classification by MRI. The healing status was assessed by checking continuity between the bone bed and meniscus on post-operative 1-year follow-up MRI scans.

### 2.3. Surgical Technique for Pullout Fixation

All surgical procedures were performed by one surgeon (K.S.C.) and involved transtibial pull-out repair using modified Mason–Allen stitches [30]. When MMPRT was confirmed on arthroscopic examination, the superficial medial collateral ligament (sMCL) was released by periosteal stripping on the distal attachment area of the sMCL. This created a sufficient working space depending on the patient’s status [31]. A meniscus resector and shaver (Linvatec; Largo, FL, USA) was used to remove the fibrous tissue and obtain fresh meniscal tissue. A curette (Linvatec; Largo, FL, USA) was then inserted through the anteromedial (AM) portal, forming a bone bed at the native root insertion site. A suture hook (Linvatec; Largo, FL, USA) loaded with No. 1 polydioxanone (PDS; Ethicon; Somerville, NJ, USA) was then passed and PDS inserted at a point 5 mm medial to the torn edge in a vertical direction. One more PDS was placed in a position inside that of the first suture, in an identical manner. The shuttle relay method completed the horizontal loop. One or two simple vertical stitches were made which overlayed and crossed the horizontal suture. After making a suitable bone tunnel by anterior cruciate ligament reconstruction tibial tunnel guide (Linvatec; Largo, FL, USA), the ends of the sutures were pulled through the tibial tunnel. Finally, the meniscus was reduced and stabilized by tying the suture over an Endo-button (Smith & Nephew; Andover, MA, USA).

### 2.4. Postoperative Rehabilitation

After three weeks of immobilization, range of motion (ROM) exercises were started and progressed to up to 90° flexion until 6-weeks after surgery. Toe touch weight-bearing using crutches commenced immediately after surgery, with the brace locked to allow for full extension of the knee joint for the first 3-weeks after surgery. Progressive partial weight-bearing exercises commenced 3-weeks following surgery. Full weight-bearing and progressive closed kinetic chain strengthening exercises were permitted 6-weeks after surgery. Light running was permitted after 3-months and sports participation after 6-months. Permanent lifestyle modifications aimed at avoiding deep knee flexion were recommended

### 2.5. Clinical Outcomes

The Lysholm score, Western Ontario and McMaster Universities Osteoarthritis Index (WOMAC) and Knee injury and Osteoarthritis Outcome Score (KOOS) were evaluated as the clinical outcomes both prior to and 1-year after surgery.

### 2.6. Radiological Outcomes

The Rosenberg 45° posteroanterior standing view was used to assess the K-L arthritis grade and measure the medial joint space both prior to surgery and 1-year after surgery [32]. The K-L grade (0/1/2/3/4) was defined as follows: grade 0 indicated no degenerative change. Grade 1 indicated questionable osteophytes and no joint space narrowing. Grade 2 showed definite osteophytes with possible joint space narrowing. Grade 3 showed definite joint space narrowing with moderate multiple osteophytes and some sclerosis. Finally, grade 4 indicated severe joint space narrowing with cysts, osteophytes and sclerosis [33]. The medial joint space was measured from the center of the medial femoral condyle to the center of the medial tibial plateau using a picture-archiving and communication system (PACS, Marotech; Infinitti, Seoul, Korea).

Each MRI scan was checked and the BME lesion, cartilage grade, healing status and meniscal extrusion were evaluated both prior to surgery and 1-year following surgery. Extrusion of the medial meniscus (in mm) was defined as the amount of meniscal displacement from the superomedial aspect of the tibial plateau to the periphery of the meniscal body at the level of the medial collateral ligament (MCL) in the coronal plane.

Radiographic images were examined independently by two authors who were blinded to the procedures used, in consultation with a single experienced musculoskeletal radiologist. All radiographic measurements were documented three times at 2-week intervals using PACS. The averages of these measurements were used in our analyses. Intraclass correlation coefficients (ICCs) were calculated to determine the interobserver reliability of the differences in radiological outcome measurements. All measurements that allowed 1 decimal value were documented three times at 2-week intervals to assess their test–retest reliability.

### 2.7. Statistical Analysis

Statistical analyses were performed using SPSS software (ver. 20.0 for Windows; SPSS Inc., Chicago, IL, USA). Statistical significance was set at 5% (*p* < 0.05). The final clinical outcomes, clinical scores and radiological outcomes for each group were compared. The Kruskal–Wallis test was used to compare variables which were not normally distributed between groups and other non-parametric values. The chi-squared (χ^2^) test was used to compare categorical data. If more than 20% of the expected frequencies were >5, Fisher’s exact test was used.

The statistical power of the study was calculated retrospectively at 79% to compare the rate of progression of the cartilage grade between group 1 (25%) and group 3 (100%) at the 0.05 significance level using Fisher’s exact test.

## 3. Results

In total, 30 patients were enrolled. The mean age of the patients was 58.7 ± 5.7 years (Table 1). Demographics and clinical characteristics prior to surgery are shown in Table 1. This was similar between all three groups, with no significant differences (*p* > 0.05).

The results of subchondral BME lesions after surgery are described in Table 2. A total of 20 patients had no change of BME lesion, 66.7% (group one). There were seven patients with improved BME lesion, 23.3% (group two). Finally, there were three patients with worsened BME lesions, 10% (group three). All ICCs ranged from 0.90 to 0.96, indicating an excellent reliability (ICCs > 0.9) of the current study [34,35]. The outcomes both before and after surgery in each group are shown in Table 3. All clinical scores following surgery were significantly improved in all groups compared with the pre-operative scores (*p* < 0.001).

When comparing the post-operative clinical and radiological outcomes between groups, the overall clinical and radiological outcomes showed a tendency for group two to be higher than other groups (Table 4). Clinically, the Lysholm score of group three was significantly lower than those of the other groups. Regarding cartilage status, the number of patients with progression of cartilage grade after surgery was five (25%) in group one, zero (0%) in group two and three (100%) in group three, showing group three to have the worst cartilage grades following surgery.

In terms of meniscal extrusion, there was no significant difference in the results in group one. Meniscal extrusion was significantly reduced in group two, whereas extrusion significantly progressed in group three (Table 3 and Table 4). Thus, a decrease in meniscal extrusion can have a positive effect on BME lesions after root repair in MMPRTs.

## 4. Discussion

The study showed 90% of the included participants to have either no change in BME lesions or an improvement of BME lesions. Only 10% of participants had worsened BME lesions. Patients with worsened BME lesions showed less favorable outcomes than other patients. Patients with improved BME lesions showed more reduced extrusion after surgery, whereas patients with worsened BME lesions showed more progressed extrusion.

BME lesions are known to be associated with the progression of osteoarthritis and a potential risk factor for structural deterioration in knee osteoarthritis [1,2,3]. BME lesions can fluctuate in size within a short time, and are associated with the progression of arthritic changes and fluctuations of pain in knee OA [1,2,3]. Thus, non-functioning meniscus by loss of hoop tension can lead to progression of BME lesions on the tibiofemoral joint.

MMPRTs are defined as an avulsion injury or radial tear occurring in the posterior bony attachment [11,12,13], and they can lead to arthritic changes due to the loss of meniscal hoop tension and load-sharing ability on the tibiofemoral joint [14]. There are presently two surgical options for MMPRT: meniscectomy [26,36] and root repair [22,23,24,25,26,27]. The ‘traditional’ gold standard treatment for MMPRT is meniscectomy, which is widely used. However, meniscectomy cannot restore meniscal hoop tension; thus, most meniscectomized cases ultimately progress to degenerative arthritis [36,37,38]. However, this study shows that better outcomes are seen after meniscal repair compared with partial meniscectomy for MMRTs. This is demonstrated through greater improvements in Lysholm scores, lower rates of progression to knee OA and a lower re-operation rate [39]. In a recently published long-term study comparing meniscectomy and root repair surgery, root repair surgery was more effective than meniscectomy in terms of both clinical outcomes and survival rate. In terms of clinical failure, defined as conversion to TKA, the 10-year survival rate of meniscectomy patients was 44.4%, whereas that of root repair surgery patients was 79.6% [21]. Thus, management with restoring meniscal hoop tension is critical to delay arthritis in MMPRTs. Moreover, it can prevent further progression of both newly formed BME lesions or existing BME lesions in the tibiofemoral joint. However, to our knowledge, there is little existing evidence to clarify the treatment’s effects of root repair through the investigation of BME lesion change following surgery.

In the current study, only three patients (10%) showed worsened BME lesions after surgery; thus, it is assumed that in almost all included participants, root repair surgery restored the meniscal hoop tension and load-bearing ability on tibiofemoral joints. This suggests root repair to be a favorable surgical procedure to prevent or delay the progression of osteoarthritis in MMPRTs, an aspect of BME lesions.

In this study, all clinical scores following surgery were significantly improved when compared with the pre-operative scores in all groups, even in worsened BME lesion patients. This suggests that root repair has better clinical outcomes following surgery in MMPRTs patients. Moreover, the overall clinical and radiological outcomes showed a tendency for group two to score higher than groups one and three. However, the Lysholm score was significantly lower in the worsened BME lesion group, group three. Radiologically, regarding cartilage status, the number of patients with a progression of cartilage grade after surgery was five patients (25%) in group one. In group two, this was 0 (0%). However, all patients with worsened BME lesions showed a worsened cartilage status after surgery, suggestive of a link between cartilage injury and BME lesions. In a previously reported study, Outerbridge grade ≥ three chondral lesions were found to be independent negative prognostic factors following root repair [23]. This suggests worsened BME lesions may be negative prognostic factors.

The results of the current study show worsened BME lesions to have a clinically unfavorable effect following root repair surgery. In MMPRTs, it is important to restore well-functioned meniscus by root repair. This study indicates that one of the goals of root repair surgery is to prevent BME lesions as much as feasibly possible. This is the clinical relevance of the study.

To our knowledge, there are no studies that have investigated the correlation between meniscus extrusion and the BME lesions. It would be worthwhile to note that this is the first study to investigate the correlation between meniscal extrusion and BME lesions. In this study, patients with improved BME lesions showed more reduced extrusion after surgery, whereas patients with worsened BME lesion showed more progressed extrusion. This means that reduced meniscal extrusion correlates with more favorable clinical outcomes compared with more progressed extrusion. Thus, more reduced meniscal extrusion after root repair can have a positive effect on BME lesions. Chung et al. reported that patients with a decreased meniscus extrusion at 1-year post-operative have more favorable clinical scores and radiographic findings at midterm follow-ups than those with increased extrusion at 1-year post-operative [40]. Meniscal extrusion has a critical impact on the results; thus, surgeons should reduce meniscus extrusion as much as possible when performing root repair. However, this result might be caused by the small sample size of this study. In the future, a larger cohort study will be needed to identify the role of meniscal extrusion in BME lesions.

This study has several limitations. First, it is a retrospective and non-randomized study; thus, there may be selection bias in the selection of participant enrollment. However, basic demographic and preoperative data were similar between groups. Therefore, any selection bias would not have any significant influence on BME lesions following surgery. Secondly, the present study performed was an in vivo study; thus, it has the potential weaknesses of an in silico study [41]. Thirdly, the sample size was relatively small, and the number of patients with worsened BME lesions was less than those of the other two groups. However, a good statistical power was achieved to compare the rates of progression of cartilage grade between groups one and three. Fourth, the follow-up duration was short at only 1-year. Finally, although the meniscal healing status was assessed by evaluating continuity between the bone bed and meniscus proper, the actual restoration of hoop tension and the healing status of the fixed meniscus was not assessed. This was due to patients not consenting to further arthroscopies.

## 5. Conclusions

Patients with worsened BME lesions have poor clinical outcomes compared with other patients. A decrease in meniscal extrusion can have a positive effect on BME lesions after root repair. Surgeons should reduce meniscus extrusion as much as possible when performing root repair. In the future, a larger prospective cohort study will be needed to identify the relevant role of meniscal extrusion in BME lesions.

## Figures and Tables

**Figure 1 medicina-58-01601-f001:**
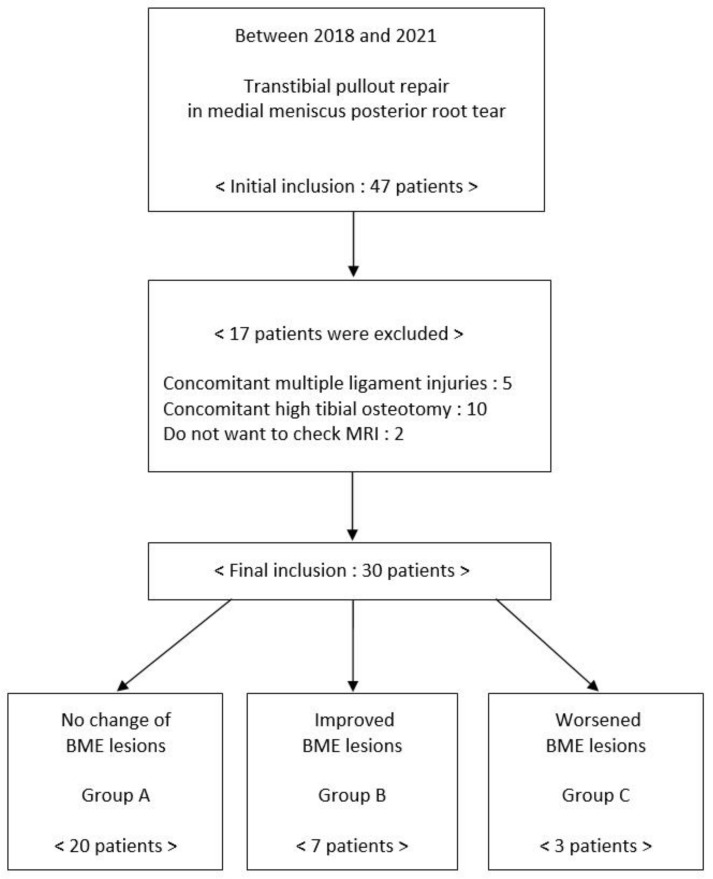
Flow chart of included participants.

**Figure 2 medicina-58-01601-f002:**
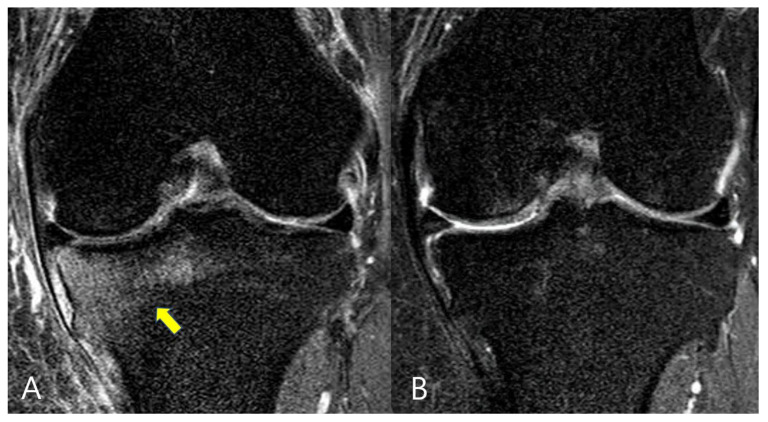
Magnetic resonance imaging (MRI) scans of a patient with a medial meniscus posterior root tear. (**A**) Pre-operative MRI view with a bone marrow edema lesion on the medial tibial plateau (yellow arrow). (**B**) One-year follow-up MRI view showing no bone marrow edema lesion on the medial tibial plateau.

**Figure 3 medicina-58-01601-f003:**
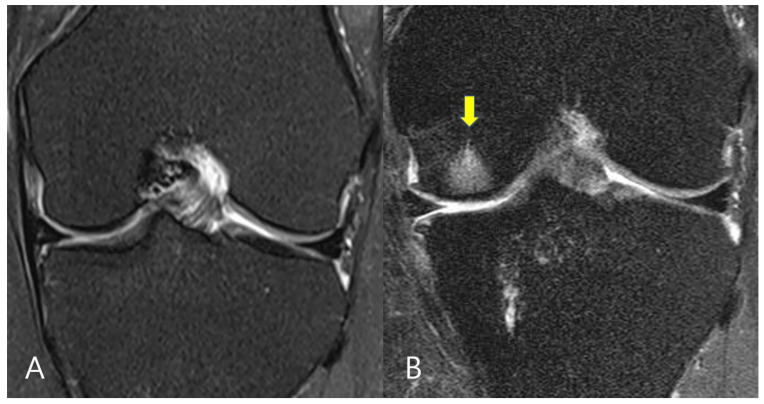
Magnetic resonance imaging (MRI) scans of a patient with a medial meniscus posterior root tear. (**A**) Pre-operative MRI view with no bone marrow edema lesion on the medial femoral condyle. (**B**) One-year follow-up MRI view showing a bone marrow edema lesion on the medial femoral condyle (yellow arrow).

**Table 1 medicina-58-01601-t001:** Pre-operative Demographics and Characteristics of Patients Included.

Demography (*n* = 30)	
Age, years	58.7 ± 5.7
Sex, male/female	2/27
Follow-up period, mo	12.2 ± 1.1
Body mass index, kg/m^2^	26.6 ± 3.9
Duration between symptom onset & repair ^a^, weeks	4.9 ± 3.6
Clinical scores	
Lysholm score	47.8 ± 6.5
WOMAC score	54.6 ± 6.7
KOOS score	69.6 ± 6.1
Radiological status	
Mechanical axis (varus), degree	3.1 ± 2.0
Kellgren–Lawrence grade (0/1/2/3/4) ^b^, *n*	518/7/0/0
Medial joint space width ^b^, mm	3.6 ± 0.6
Outerbridge cartilage grade (0/1/2/3/4) ^c^, *n*	5/6/13/5/1

^a^ The point of diagnosis was defined as the time when painful symptoms originally occurred. ^b^ The values were acquired through posteroanterior 45° flexion weight-bearing radiographs. ^c^ The values were acquired through magnetic resonance imaging. Data are presented as mean ± standard deviation.

**Table 2 medicina-58-01601-t002:** Bone Marrow Lesions (BME) of Included Participants.

BME Lesion (*n* = 30)			
	Pre-Operation (*n*)	Change of BME Lesions of Included Participants at Final Follow-Up (*n*)	Post-Operation (*n*)
No BME lesion	20	No BME: 17 BME on MFC: 1 BME on MTP: 1 BME on both: 1	23
BME lesion on MFC only	1	BME on MFC: 1	3
BME lesion on MTP only	6	No BME: 6	1
BME lesion on MFC and MTP both	3	BME on MFC: 1 BME on both: 2	3

MFC: medial femoral condyle. MTP: medial tibial plateau.

**Table 3 medicina-58-01601-t003:** Pre-operative and Post-operative Clinical and Radiological Outcomes in Each Group.

BME Lesion (*n* = 29)						
	No Change of BME Lesion (Group 1)		Improved BME Lesion (Group 2)		Worsened BME Lesion (Group 3)	
Number	20		7		3	
Clinical scores	Preoperation	Postoperation	Preoperation	Postoperation	Preoperation	Postoperation
Lysholm score	47.6 ± 6.8	81.7 ± 8.2	50.1 ± 6.3	86.9 ± 6.7	44.0 ± 3.5	73.7 ± 4.0
WOMAC score	53.1 ± 7.2	16.7 ± 8.7	57.3 ± 4.3	12.7 ± 4.5	58.7 ± 4.0	22.3 ± 6.8
KOOS score	68.8 ± 6.5	26.5 ± 9.6	69.7 ± 4.6	24.1 ± 5.8	75.0 ± 3.6	34.3 ± 10.7
Radiological status						
Kellgren–Lawrence grade (0/1/2/3/4) ^a^, *n*	4/12/4/0/0	2/11/5/2/0	0/5/2/0/0	0/4/3/0/0	1/1/1/0/0	1/0/2/0/0
Medial joint space width ^a^, mm	3.6 ± 0.6	3.3 ± 0.5	3.5 ± 0.8	3.4 ± 0.8	3.4 ± 0.5	3.0 ± 0.3
Meniscus extrusion ^b^, mm	3.4 ± 0.8	3.3 ± 1.1	3.7 ± 1.0	2.7 ± 0.8	2.8 ± 0.9	5.5 ± 1.5
Outerbridge cartilage grade (0/1/2/3/4) ^b^, *n*	4/4/10/2/0	3/4/9/4/0	0/1/3/2/1	2/2/2/1/0	1/1/0/1/0	0/1/1/0/1

^a^ The values were acquired through posteroanterior 45o flexion weight-bearing radiographs. ^b^ The values were acquired through magnetic resonance imaging. Data are presented as mean ± standard deviation.

**Table 4 medicina-58-01601-t004:** Postoperative Clinical and Radiological Outcomes Between Groups ^a^.

Postoperative Outcomes	No Change of BME Lesion (*n* = 20)	Improved BME Lesion (*n* = 7)	Worsened BME Lesion (*n* = 3)	*p* Value
Clinical scores				
Lysholm score	81.7 ± 8.2	86.9 ± 6.7	73.7 ± 4.0	0.034 ^b^
Difference of Lysholm between pre- and post-operation	34.1 ± 8.2	36.7 ± 11.2	29.7 ± 1.5	0.416 ^b^
WOMAC score	16.7 ± 8.7	12.7 ± 4.5	22.3 ± 6.8	0.160 ^b^
Difference of WOMAC between pre- and post-operation	36.4 ± 12.7	44.6 ± 2.7	36.3 ± 2.9	0.106 ^b^
KOOS score	26.5 ± 9.6	24.1 ± 5.8	34.3 ± 10.7	0.291 ^b^
Difference of KOOS between pre- and post-operation	42.3 ± 9.5	45.6 ± 8.0	40.7 ± 10.3	0.431 ^b^
Radiological status				
Medial joint space, mm	3.3 ± 0.5	3.4 ± 0.8	3.0 ± 0.3	0.623 ^b^
Progression of joint space narrowing, mm	0.4 ± 0.4	0.1 ± 0.2	0.4 ± 0.2	0.081 ^b^
Kellgren–Lawrence grade (KL grade), 0/1/2/3/4	2/11/5/2/0	0/4/3/0/0	1/0/2/0/0	0.347 ^c^
Progression of KL grade, no. (%)	7 (35%)	1 (14%)	1 (33%)	0.829 ^c^
Continuity between bone bed and meniscus ^d^	20 (100%)	7 (100%)	3 (100%)	1.000 ^c^
Difference values of meniscus extrusion (pre-post) ^d^, mm	0.1 ± 1.3	0.9 ± 1.4	−2.7 ± 1.5	0.019 ^b^
Cartilage grade (modified outerbridge), 0/1/2/3/4 ^d^	3/4/9/4/0	2/2/2/1/0	0/1/1/0/1	0.548 ^c^
Progression of Cartilage grade, no. (%) ^d^	5 (25%)	0 (0%)	3 (100%)	0.006 ^c^

^a^ Values are expressed as mean ± standard deviation. ^b^ Kruskal–Wallis test. ^c^ Fisher exact test. ^d^ The value was taken from magnetic resonance imaging checked at postoperative 1-year follow-up.

## Data Availability

The data presented in this study are available on request from the corresponding author.

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
