# Peer review of "Changes in Bone Marrow Lesions Following Root Repair Surgery Using Modified Mason–Allen Stitches in Medial Meniscus Posterior Root Tears"

_medicina, 2022, doi:10.3390/medicina58111601_

Round 1

Reviewer 1 Report

1.      It is needed to provide all of the Author’s email after affiliation (with initials author name if more than 1) following MDPI format, except for corresponding.

2.      As the conclusion of your abstract, please provide a "take-home" message.

3.      Keywords should have been reorganized alphabetically.

4.      Please use lowercase font for each of the keywords in accordance with MDPI format.

5.      The Reviewer do not see the novel in the present article. My examination revealed that several similar previous publications appear to appropriately address the issues you have brought up in the current submission. Please emphasize it more advance in the introduction section if there are any more truly something really new.

6.      Previous study related needs to explain in the introduction section consisting of their work, their novelty, and their limitations to show the research gaps that intend to be filled in the present study.

7.      Since the present study performs in vivo study, it would discuss the potential of in silico study. It is a vital topic that authors must provide in the introduction and/or discussion section. Additionally, the MDPI's suggested reverence should be taken to substantiate this explanation as follows: Jamari, J.; Ammarullah, M. I.; Santoso, G.; Sugiharto, S.; Supriyono, T.; Heide, E. van der. In Silico Contact Pressure of Metal-on-Metal Total Hip Implant with Different Materials Subjected to Gait Loading. Metals (Basel). 2022, 12, 1241. https://doi.org/10.3390/met12081241

8.      To enhance the understandability of the section on materials and methods easier for them to understand rather than just depending on the main text as it exists at the moment, the authors could add additional illustrations in the form of figures that explain the workflow of the present study.

9.      What is the basis for rpatient selection? Is there any protocol, standard, or basis that has been followed? It is unclear since the patient is very heterogeneous with a small number. The resonance involved impacts the present result makes this study flaws. One major reason for rejecting this paper.

10.   It is necessary to provide more information on the manufacturer, country, and specifications of the tools.

11.   A comparative assessment with similar previous research is required.

12.   Please explain the further research in the conclusion section.

13.   Five years back literature should be enriched into the reference, and MDPI-published literature is highly recommended.

14.   Throughout the whole manuscript, the authors sometimes wrote paragraphs with just one or two phrases, which made the explanation difficult to understand. To make their explanation a full paragraph, the writers should expand it. It is advised to use at least three sentences in a paragraph, with the primary sentence coming first and the supporting sentences coming after.

15.   Due to grammatical problems and linguistic style, the authors should proofread the work. It would be used MDPI English editing service for this concern.

16.   Ensure that the authors followed the MDPI format exactly, edit the current form, and double-check all of the previously noted problems.

Author Response

Dear Editor in Chief & Editors & Reviewers 

Thank you for your faithful reviews. We wish to re-submit the attached manuscript as an Original Article. The manuscript has been rechecked and appropriate changes have been made in accordance with the reviewers’ suggestions. The responses to their comments have been prepared and attached herewith. Changes made in accordance with the reviewer comments are in track change.

We thank you and the reviewers for your thoughtful suggestions and insights, which have enriched the manuscript and produced a better and more balanced account of the research.

Please, your kind consideration of this paper would be greatly appreciated. I hope you will consider this article as suitable for publication in Medicina. We are looking forward to your answer.

Thank you very much.

Sincerely,

----------------------------------------------------------------------------------------------------------------------

Reviewer 1

Comments and Suggestions for Authors

  1. It is needed to provide all of the Author’s email after affiliation (with initials author name if more than 1) following MDPI format, except for corresponding.
  • Thank you for your comment. We added authors’ email after affiliation. Please see affiliation part in revised manuscript (red pont).
  1. As the conclusion of your abstract, please provide a "take-home" message.
  • Thank you for your comment. We modified the conclusion of abstract (focusing meniscus extrusion by other reviewer’s recommendation). Please see the abstract (track change). Please take the review into kind consideration.
  1. Keywords should have been reorganized alphabetically.
  • Thank you for your comment. We modified the Keywords. Please see the Keywords part (Track change).
  1. Please use lowercase font for each of the keywords in accordance with MDPI format.
  • Thank you for your comment. We modified the Keywords. Please see the Keywords part (Track change).
  1. The Reviewer do not see the novel in the present article. My examination revealed that several similar previous publications appear to appropriately address the issues you have brought up in the current submission. Please emphasize it more advance in the introduction section if there are any more truly something really new.
  • Thank you for your comment. To our knowledge, there are no studies that have investigated the correlation between meniscus extrusion and the BME lesions. It would be worthwhile that this is the first study to investigate the correlation between meniscal extrusion and BME lesions. In this study, patients with improved BME lesion showed more reduced extrusion after surgery, whereas patients with worsened BME lesion showed more progressed extrusion. It means that reduced meniscal extrusion correlated with more favorable clinical outcomes compared with more progressed extrusion. Thus, more reduced meniscal extrusion after root repair can make positive effect on BME lesion. However, this result might be caused by the small sample size of this study. In the future, a larger cohort study will be needed to identify the role of meniscal extrusion in BME lesions.

We added this information in the Abstract (track change) and Line 233-247 (discussion part, track change) in revised manuscript. Please take the review into kind consideration.

  1. Previous study related needs to explain in the introduction section consisting of their work, their novelty, and their limitations to show the research gaps that intend to be filled in the present study.
  • Thank you for your comment. To our knowledge, there are no studies that have investigated the correlation between meniscus extrusion and the BME lesions. It would be worthwhile that this is the first study to investigate the correlation between meniscal extrusion and BME lesions. In this study, patients with improved BME lesion showed more reduced extrusion after surgery, whereas patients with worsened BME lesion showed more progressed extrusion. It means that reduced meniscal extrusion correlated with more favorable clinical outcomes compared with more progressed extrusion. Thus, more reduced meniscal extrusion after root repair can make positive effect on BME lesion. This is novelity of this study. However, this result might be caused by the small sample size of this study. In the future, a larger cohort study will be needed to identify the role of meniscal extrusion in BME lesions.

Please see Line 26-38 (introduction, track change), the Abstract (track change) and Line 233-247 (discussion part, track change) in revised manuscript. Please take the review into kind consideration.

  1. Since the present study performs in vivo study, it would discuss the potential of in silico study. It is a vital topic that authors must provide in the introduction and/or discussion section. Additionally, the MDPI's suggested reverence should be taken to substantiate this explanation as follows: Jamari, J.; Ammarullah, M. I.; Santoso, G.; Sugiharto, S.; Supriyono, T.; Heide, E. van der. In Silico Contact Pressure of Metal-on-Metal Total Hip Implant with Different Materials Subjected to Gait Loading. Metals (Basel). 2022, 12, 1241. https://doi.org/10.3390/met12081241
  • Thank you for your comment. We added in Line 252-253 (track change) in revised manuscript (reference 41). Please take the review into kind consideration.
  1. To enhance the understandability of the section on materials and methods easier for them to understand rather than just depending on the main text as it exists at the moment, the authors could add additional illustrations in the form of figures that explain the workflow of the present study.
  • Thank you for your comment. We included flaw chart of participants as figure 1. However, we cannot understand how to add illustrations in the form of figures that explain the workflow of the present study. Thus, we politely ask if you let us know how to do thorough an example, we will add it.
  1. What is the basis for patient selection? Is there any protocol, standard, or basis that has been followed? It is unclear since the patient is very heterogeneous with a small number. The resonance involved impacts the present result makes this study flaws. One major reason for rejecting this paper.
  • Thank you for your comment. Inclusion criteria for the study were: first, MMPRT shown on MRI in a patient with persistent knee pain; second, patients who underwent arthroscopic pull-out fixation by modified Mason-Allen stitches; third, patients with a Kellgren-Lawrence (K-L) of grade two or less. Study exclusion criteria were: first, patients whose MMPRT was combined with a high tibial osteotomy; second, patients with a concomitant ligament injury; and finally, patients who did not want MRI scan follow-up 1-year after surgery (Figure 1).

However, this study is a retrospective and non-randomized study with small number participants (not prospective randomized study), thus, there may be selection bias in the selection of participant enrollment. We added this problem in the Limitation (Please see 248-249, 253-256). Please take the review into kind consideration.

  1. It is necessary to provide more information on the manufacturer, country, and specifications of the tools.
  • Thank you for your comment. We add more information in Line 94, 95 (track change). Please take the review into kind consideration.
  1. A comparative assessment with similar previous research is required.
  • Thank you for your comment. We added one study which investigated the correlation between meniscal extrusion and midterm clinical outcomes (reference 40). Chung et al. reported that patients with decreased meniscus extrusion at postoperative 1 year have more favorable clinical scores and radiographic findings at midterm follow-up than those with increased extrusion at 1 year. However, that study (reference 40) did not investigate the correlation between meniscus extrusion and BME lesions.

To our knowledge, there are no studies that have investigated the correlation between meniscus extrusion and the BME lesions. It would be worthwhile that this is the first study to investigate the correlation between meniscal extrusion and BME lesions. In this study, patients with improved BME lesion showed more reduced extrusion after surgery, whereas patients with worsened BME lesion showed more progressed extrusion. It means that reduced meniscal extrusion correlated with more favorable clinical outcomes compared with more progressed extrusion. Thus, more reduced meniscal extrusion after root repair can make positive effect on BME lesion. Meniscal extrusion has a critical impact on the results, thus, surgeons should reduce meniscus extrusion as much as possible when performing root repair. However, this result might be caused by the small sample size of this study. In the future, a larger cohort study will be needed to identify the role of meniscal extrusion in BME lesions.

Please see Line 233-247 in revised manuscript. Please take the review into kind consideration.

  1. Please explain the further research in the conclusion section.
  • Thank you for your comment. We added your recommendation in the conclusion section. Please see Line 266-269 (track change).
  1. Five years back literature should be enriched into the reference, and MDPI-published literature is highly recommended.
  • Thank you for your comment. We tried to insert a proper reference. Please kindly understand that it was difficult to insert proper references published in MDPI journals because the field of root repair was rarely published in the MDPI journals. Please take the review into kind consideration.
  1. Throughout the whole manuscript, the authors sometimes wrote paragraphs with just one or two phrases, which made the explanation difficult to understand. To make their explanation a full paragraph, the writers should expand it. It is advised to use at least three sentences in a paragraph, with the primary sentence coming first and the supporting sentences coming after.
  • Thank you for your comment. We tried to modify according to your recommendation. Please take the review into kind consideration.
  1. Due to grammatical problems and linguistic style, the authors should proofread the work. It would be used MDPI English editing service for this concern.
  • Thank you for your comment. This manuscript was corrected by native speaker. Before final publish, we will take English correction service again.
  1. Ensure that the authors followed the MDPI format exactly, edit the current form, and double-check all of the previously noted problems.
  • Thank you for your comment. We tried to follow the MDPI format exactly.

Finally, we sincerely thank for your thoughtful suggestions and insights, which have enriched the manuscript and produced a better and more balanced account of the research. Again, thank you very much for your nice comment.

Reviewer 2 Report

Changes in Bone Marrow Lesions Following Root Re-pair Surgery using Modified Mason-Allen Stitches in Medial Meniscus Posterior Root Tears

Concerns: 
Conclusion based on the described data is not allowed. There is no control group and the improvement regarding bone edema is not really demonstrated by the data (66.7% w/o changes), that means the majority.
The repair success of the meniscus and the amount of extrusion is not analyzed and shown in the postoperative MRI's.

The study do not improve the scientific knowledge regarding meniscal root tears.

Suggestions: Describe the preoperative versus the postoperative extrusion of the meniscus and align it with the bone edema. Do not make conclusions like bone edema is improved with these results and w/o a control group.

Emphasis on description of the results

Author Response

Dear Editor in Chief & Editors & Reviewers 

Thank you for your faithful reviews. We wish to re-submit the attached manuscript as an Original Article. The manuscript has been rechecked and appropriate changes have been made in accordance with the reviewers’ suggestions. The responses to their comments have been prepared and attached herewith. Changes made in accordance with the reviewer comments are in track change.

We thank you and the reviewers for your thoughtful suggestions and insights, which have enriched the manuscript and produced a better and more balanced account of the research.

Please, your kind consideration of this paper would be greatly appreciated. I hope you will consider this article as suitable for publication in Medicina. We are looking forward to your answer.

Thank you very much.

Sincerely,

Review 2

Concerns: 
Conclusion based on the described data is not allowed. There is no control group and the improvement regarding bone edema is not really demonstrated by the data (66.7% w/o changes), that means the majority.

  • Thank you for your comment. We respect your opinion. Since we recommended root repair as soon as possible (less than 2-3 weeks after diagnosis), there were not many cases of BME lesion. BME lesions are a risk factor for structural deterioration in knee osteoarthritis and strongly associated with the progression of osteoarthritis. Thus, if root repair is not effective or successful, BME lesions will worsen or progress following surgery. The fact that the edema did not deteriorate after the operation is considered to be effectively maintained. Please take the review into kind consideration.

The repair success of the meniscus and the amount of extrusion is not analyzed and shown in the postoperative MRI's.

  • Thank you for your comment. We added meniscal extrusion data in Line 133-135 (methods, track change), Table 3 and 4 (results, red pont), Line 174-177 (results, track change), Line 233-247 (discussion, red pont) in revised manuscript. Please take the review into kind consideration.

The study do not improve the scientific knowledge regarding meniscal root tears.

  • Thank you for your comment. We modified according to your suggestions. Please take the review into kind consideration.

Suggestions: Describe the preoperative versus the postoperative extrusion of the meniscus and align it with the bone edema.

  • Thank you for your comment. We added meniscal extrusion data in Line 133-135 (methods, track change), Table 3 and 4 (results, red pont), Line 174-177 (results, track change), Line 233-247 (discussion, red pont), abstract (track change) in revised manuscript. Please take the review into kind consideration.

Do not make conclusions like bone edema is improved with these results and w/o a control group. Emphasis on description of the results

  • Thank you for your comment. We modified conclusion in revised manuscript. Please see abstract (track change) and Line 266-269 (track change). Please take the review into kind consideration.

Finally, we sincerely thank for your thoughtful suggestions and insights, which have enriched the manuscript and produced a better and more balanced account of the research. Again, thank you very much for your nice comment.
